# Miro GTPases at the Crossroads of Cytoskeletal Dynamics and Mitochondrial Trafficking

**DOI:** 10.3390/cells13070647

**Published:** 2024-04-07

**Authors:** Pontus Aspenström

**Affiliations:** Rudbeck Laboratory, Department of Immunology, Genetics and Pathology (IGP), Uppsala University, SE 751 85 Uppsala, Sweden; pontus.aspenstrom@igp.uu.se; Tel.: +46-18-4710000

**Keywords:** Miro GTPases, Parkinson’s disease, neuropathology, microtubules, mitochondrial dynamics

## Abstract

Miro GTPases are key components in the machinery responsible for transporting mitochondria and peroxisomes along microtubules, and also play important roles in regulating calcium homeostasis and organizing contact sites between mitochondria and the endoplasmic reticulum. Moreover, Miro GTPases have been shown to interact with proteins that actively regulate cytoskeletal organization and dynamics, suggesting that these GTPases participate in organizing cytoskeletal functions and organelle transport. Derailed mitochondrial transport is associated with neuropathological conditions such as Parkinson’s and Alzheimer’s diseases. This review explores our recent understanding of the diverse roles of Miro GTPases under cytoskeletal control, both under normal conditions and during the course of human diseases such as neuropathological disorders.

## 1. Introduction

The Miro GTPases Miro-1 and Miro-2 were originally identified more than twenty years ago, through advanced searches of DNA sequence databases generated by the human genome project [1]. The original aim of these searches was to find unidentified novel members of the Rho family of monomeric GTPases. This venture identified new Rho GTPases, as well as two Rho-related proteins with rather peculiar domain organizations [2]. These proteins were shown to harbor two GTPase-like domains: one in the N-terminus (nGTPase) and one in the C-terminus (cGTPase). Moreover, two sets of calcium-binding EF hands were identified in the linker region between these two GTPase domains (Figure 1) [1]. In addition, analysis of the three-dimensional structure of the Drosophila melanogaster Miro orthologue exposed two sets of hidden EF-hand motifs adjacent to the classical ones, although it remains unclear whether these unconventional hidden EF hands can serve as true calcium-binding motifs [3]. These motifs are also visible in the three-dimensional models of human Miro GTPases (Figure 1). Rho GTPases are known to function as key regulators of the actin cytoskeleton. However, a side-by-side analysis of novel and well-established Rho GTPases indicated Miro GTPases had no obvious impact on actin filament organization. Instead, they were found to localize to the mitochondria and were thus named Mitochondrial Rho or Miro for short [1,2,4]. Mitochondrial targeting is achieved through a C-terminal alpha helical motif inserted into the outer mitochondrial membrane [5]. Due to their mitochondrial localization and the fact that Miro GTPases lack the insert domain present in all classical Rho GTPases, Miro GTPases are now considered a separate subfamily within the Ras superfamily of monomeric GTPases [6].

Since their first characterization, Miro GTPases have been shown to play key roles in regulating mitochondrial dynamics, predominantly serving as adaptors in the protein complex that regulate the trafficking of mitochondria along microtubule tracks. In addition, Miro GTPases have been implicated in Parkinson’s disease—and possibly other neuropathological conditions—based on their interactions with the Parkinson protein Parkin, which is an E3 ubiquitin ligase, and the PTEN-induced kinase (PINK1), which is a Ser/Thr protein kinase [7,8,9,10]. This article only briefly discusses these well-established roles of Miro GTPases and, instead, focuses on the emerging understanding that Miro GTPases can play more active roles in regulating cytoskeletal dynamics and how defects in this regulation can be linked to human diseases.

**Figure 1 cells-13-00647-f001:**
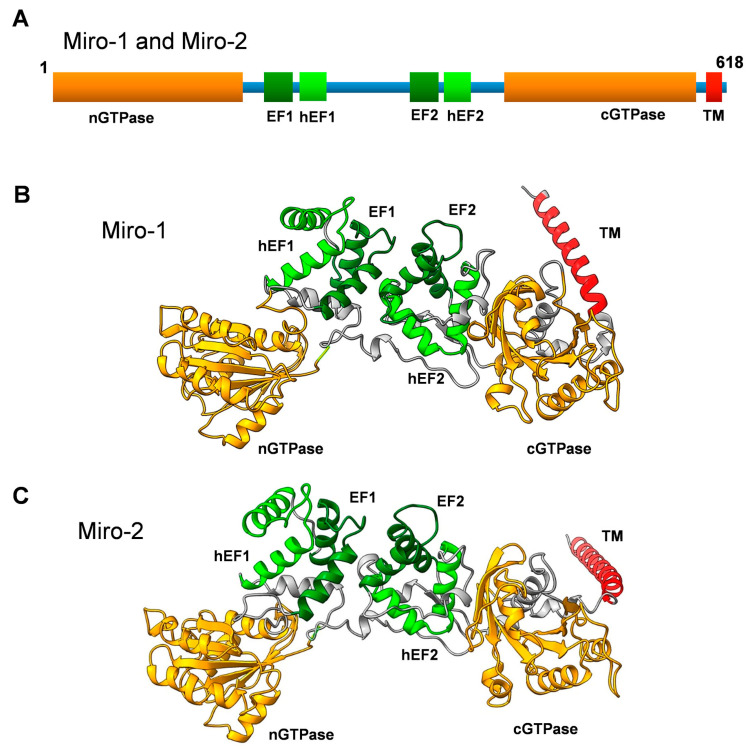
(**A**) Primary structure of Miro-1 and Miro-2 showing a schematic representation of the domain organization. nGTPase and cGTPase denote the N-terminal and C-terminal GTPase domains, respectively. hEF1 and hEF2 denote the hidden EF-like motifs, and EF1 and EF2 denote the classical EF-hand motifs. TM denotes the transmembrane motif. (**B**) Three-dimensional structure model of human Miro-1 created with the AlphaFold 3D model prediction software [11,12]. Note that the TM motif is not well structured in the model and might have a completely different orientation. (**C**) Three-dimensional structural model of human Miro-2 created with the AlphaFold 3D model prediction software [11,12]. Note that the TM motif is not well structured in the model and might have a completely different orientation.

## 2. Domain Organization of Miro GTPases

Miro GTPases are evolutionarily conserved throughout the eukaryotic kingdom. Unicellular eukaryotes, such as Saccharomyces cerevisiae and Schizosaccharomyces pombe, each have one gene called GEM1 that encodes Miro GTPases [1,13]. This structure is also found in smaller eukaryotes such as Dictyostelium discoideum (GemA), Drosophila melanogaster (dMiro), and Caenorhabditis elegans (miro-1) [1,14,15]. There are two additional Miro-like genes in C. Elegans called miro-2 and miro-3, which are unlikely to be functional [15]. Most higher eukaryotes contain two Miro genes, but zebrafish and plants such as Arabidopsis thaliana have three genes encoding Miro orthologues, while Physcomitrella patens (Spreading Earth-moss) has four [14]. The human genome contains two genes encoding Miro proteins: RHOT1, which encodes Miro-1, and RHOT2, which encodes Miro-2. Throughout this text, I refer to human Miro GTPases unless stated otherwise.

Miro-1 and Miro-2 are 60% identical and almost 80% similar to each other. Both proteins constitute polypeptides of 618 amino acid residues with predicted molecular masses of 70 kDa. The nGTPase domain is most important for the functioning of Miro GTPases, as mutations disrupting the nucleotide-binding or hydrolysis activity of this domain have dramatic effects on mitochondrial localization, resulting in the accumulation of mitochondria in the perinuclear region when ectopically expressed in cultured cells [1,2,5]. Similar types of mutations in the cGTPase domain have no significant effects on mitochondrial distribution, which could imply that the nGTPase domain is the key actor in Miro’s functions. However, the adaptor protein Trafficking kinesin 1 (TRAK1), which links Miro to kinesin microtubule motors, appears to bind to a C-terminal fragment of Miro-1 involving the EF-hands and the cGTPase domain. This interaction occurs independently of the GTP/GDP-bound status of Miro-1 [16]. Thus, it is likely that the cGTPase domain functions as a binding domain for target proteins and the nGTPase domain as a regulatory domain. Although this is the predominant domain architecture of Miro GTPases throughout evolution, there are exceptions. For instance, Miro GTPases from Trypanosoma and Leishmania lack the nGTPase domain and instead harbor a unique N-terminal domain without resemblance to any known type of domain [14]. In contrast, Miro GTPases in ciliates such as Paramecium lack the cGTPase domain entirely, or the domain represents a deteriorated GTPase domain without the consensus nucleotide-binding motifs [14].

A closer examination of the two Miro GTPase domains reveals some atypical features of the GTP-binding motifs. In Ras-type classical monomeric GTPases, there are five main motifs named the G1–G5 loops, which are responsible for binding the nucleotide [17]. While the G4 and G5 loops are conserved in Miro GTPases, the G2 and G3 loops (also known as switch 1 and switch 2) are not. Moreover, the G1 loop (also known as the P-loop) deviates from classical monomeric GTPases. The position equivalent to Ras codon 12 is a glycine residue in the Ras and classical Rho GTPases, while in the Miro nGTPase domain, this position is instead occupied by a proline (Miro-1) or alanine (Miro-2). In the cGTPase domain, this position is occupied by a lysine (Miro-1) or arginine (Miro-2) [1]. There are also deviations from consensus sequences in the positions equivalent to Ras codon 61, which is a glutamine in Ras and the classical Rho GTPases. However, in Miro, alanine residues can instead be found in the nGTPase domains, and a serine (Miro-1) or an aspartic acid (Miro-2) can be found in the cGTPase domains [1]. These amino acid replacements in position 12 and 61 (Ras numbering) are found in oncogenic Ras and render Ras proteins GTPase-deficient and, therefore, constitutively active [18]. In this regard, Miro GTPases resemble GTPase-deficient atypical Rho GTPases, such as Rnd GTPases, RhoBTBs, and RhoH [19]. Paradoxically, Miro GTPases have both GTP-binding capacity and GTPase activity [20,21,22,23]. The nGTPase domain of the yeast orthologue Gem1 showed marked GTP hydrolysis activity [24]. In addition, the Drosophila nGTPase domain was found to be kinetically active [21]. The situation for the human nGTPase domain, however, is more unclear. One study detected weak enzymatic activity [20] that increased through phosphorylation by the Polo kinase [21]. The Vibrio cholerae effector VopE was also found to positively affect the GTPase activity of the nGTPase domain, which suggests that this effector could serve as a GTPase-activating protein (GAP) for Miro GTPases [22]. To date, however, no endogenous GAP proteins have been identified for Miro GTPases. It is difficult to relate these GTPase measurements to the hydrolysis activity of conventional small GTPases, as no Ras- or Rho-type monomeric GTPases were tested under the same conditions. Moreover, structural analysis of the nGTPase domain demonstrated that it binds GTP with high affinity, indicating that this domain likely has very weak GTPase activity [23]. The nGTPase domain from smaller eukaryotes may possess active hydrolysis activity as opposed to the nGTPase domain of human Miro. The apparent difference in kinetic behavior between human Miro nGTPase and the yeast and Drosophila equivalents could be related to differences in amino acid sequences in the G1 loop. The sequences of the Gem-1 and the dMiro nGTPase domains are more similar to the consensus GxGxxG motif, as they have GxxGxG rather than the Miro-1 equivalent, which completely lacks the central glycine residue [2]. All studies to date have involved the nGTPase domain of Miro-1, and it is not known if the corresponding domain of Miro-2 is functionally different. It is also unclear if the cGTPase domain can function as a catalytically active GTPase domain. One study demonstrated the weak GTPase activity of the Miro-1 cGTPase domain. Interestingly, the cGTPase domain was also catalytically active towards ATP and UTP but not CTP [20].

Miro GTPases have tandem EF hands known as EF1 and EF2. Structural analysis of Drosophila dMiro uncovered some unusual features in the EF hands of Miro GTPases. The crystal structure of a C-terminal fragment containing the two EF-hand motifs and the cGTPase domain, known as dMiroS, was determined at between 2.8 and 3.0 Å, depending on whether the structure contained Mg^2+^ and/or Ca^2+^ or lacked divalent cations altogether [3]. The EF hands follow a classical EF-hand fold with glutamate residues at a critical position to allow Ca^2+^ binding. Nevertheless, cations in the structure bound only to EF2, which suggests that only one of the EF hands is fully functional. In addition, the overall fold of dMiroS was nearly identical regardless of whether EF2 lacked the cation or Mg^2+^ or Ca^2+^ occupied the binding site. In addition, each EF hand is paired with a non-canonical hidden EF hand, hEF1 or hEF2 (Figure 1) [3]. However, the hEF hands are likely incapable of binding divalent cations. Despite these uncertainties regarding the binding mode of the EF hands, calcium binding to Miro GTPases constitutes an important regulatory switch with a direct impact on mitochondrial motility along the microtubules, as calcium release arrests mitochondrial trafficking [25,26]. How this phenomenon is achieved at the molecular level remains unclear. However, calcium binding does not appear to induce any major conformational switch to dMiroS [3].

## 3. Miro GTPases Are Involved in Organelle Homeostasis

Early observations indicated that Miro GTPases seem to play no major role in cytoskeletal regulation, despite their similarities, at least in the N-terminal GTPase domain, to classical Rho GTPases. When epitope-tagged variants of Miro-1 and Miro-2 were expressed in fibroblasts, no obvious effects on actin organization were observed [1,2,5]. This result contrasts with classical Rho GTPases, which all have profound effects on actin organization [2,27]. Instead, Miro GTPases were found to localize to mitochondria [1]. Importantly, Miro GTPases harboring mutants in a position equivalent to Ras codon 12, Miro-1P13V and Miro2A13V, presumably rendered the proteins constitutively active and resulted in an accumulation of mitochondria in the perinuclear area around the centrosome when expressed in cultured cells [1]. Notably, the presence of proline and alanine residues in position 13 of Miro-1 and Miro-2, respectively, signals that the proteins are not fully functional hydrolyzing enzymes. Thus, why these amino acid substitutions have such a dramatic effect on Miro GTPases is unclear.

As mitochondria are transported along microtubule tracks, they must have some impact on cytoskeletal function, but these effects are uncertain. The first answer came from analyses of Drosophila, which found that dMiro links mitochondria to microtubules through the adapter protein Milton and the kinesin motor protein [28]. Subsequently, human Miro GTPases were also found to interact with the human Milton orthologues TRAK1 (also known as O Glc NAc Transferase Interacting Protein 106 kDa-OIP106) and TRAK2 (also known as GABA(A) receptor interacting factor 1-GRIF1) [5,25]. However, these findings could not explain the accumulation of mitochondria in the perinuclear area. Kinesin motor proteins normally regulate transport in the anterograde direction, i.e., towards the microtubule positive ends directed towards the cell periphery. Retrograde motility is predominantly controlled by dynamin motor proteins. In agreement with this observation, dMiro was found to link both kinesin and dynein via the adapter Milton and play roles in bidirectional motility [29]. These original observations were followed by numerous publications outlining the concept of Miro GTPases in microtubule motor-dependent mitochondrial trafficking (see, e.g., [6,7,8] for recent reviews). In addition, Miro GTPases have been found to play important roles as components in ER–mitochondria contact sites and in regulating mitochondrial calcium release [30,31,32].

Mitochondria are highly flexible organelles subject to a constant process of reshaping through fusion and fission events. A number of proteins with roles in the regulation of mitochondrial fusion and fission have also been identified [33]. Mitochondrial fusions are mainly regulated by Mitofusion-1 and -2 and the optic atrophy 1 protein (OPA1). Mitochondrial fission is regulated by the Dynamin-related protein 1 (Drp1) [33]. Miro GTPases were found to interact with Mitofusin-2, which originally suggested that Miro GTPases could play a regulatory role in mitochondrial fusion [34]. Mutations in Mitofusin-2 were found in the hereditary motor and sensory neuropathy Charcot–Marie–Tooth disease (CMT) [35]. However, the expression of CMT-related mutations in Mitofusin-2 and knockdown of Mitofusin-2 in cultured neurons derailed anterograde and retrograde mitochondrial motility, rather than mitochondrial fusion defects [34]. This observation indicated that Mitofusins could also play roles in regulating mitochondrial trafficking. A slightly different result was achieved in a study using human embryonic kidney HEK293 cells [34]. In this study, it was shown that an elevated concentration of calcium inhibited Mitofusin-dependent fusion. The authors suggested that Miro-1 serves as a calcium sensor and mediates the inhibition of mitochondrial fusion in a calcium-dependent manner [36]. The interplay between Miro GTPases and Mitofusins indicated that these proteins could bind to each other. Indeed, Mitofusins were also found to have the capacity for direct interactions with Miro GTPases [34,36]. Studies of mouse embryonic fibroblasts (MTFs) isolated from Miro-1, Miro-2, or Miro-1/Miro-2 double-knockout mice suggested that Miro-1 but not Miro-2 is involved in microtubule-dependent mitochondrial motility. Surprisingly, the analysis also showed that TRAK1 and TRAK2 can be localized to mitochondria and participate in mitochondrial dynamics, indicating that Miro GTPases are not obligatory for this function. Mitofusins could potentially compensate for some of the Miro-dependent roles in mitochondria transport [37].

In terms of mitochondrial fission, expression of the dominant negative mutants Miro-1/T18N in combination with Miro-2/T18N or the silencing of Miro-1 resulted in mitochondrial fragmentation in rat cardiomyocyte H9c2 cells [25]. This result indicates communication between Miro GTPases and the mitochondrial fission machinery. However, it remains unclear if this occurs through direct interactions. One study found that collapsin response mediator protein 2 (CRMP2) interacts with Drp1 and Miro-2 to regulate mitochondrial fission in human neurons [38]. Another example of communication between Miro GTPases and the fission regulator Drp1 was observed in the regulation of peroxisome dynamics. Peroxisomes play important roles in lipid metabolism and the reduction of reactive oxygen species, with their sizes, shapes, and numbers tightly controlled [39]. Miro GTPases are involved in the Drp1-mediated maintenance of peroxisomal size through a mechanism involving inhibitory effects on peroxisomal Drp1 [40]. Finally, Miro GTPases were found to regulate the transport of mitochondria between cells through tunneling nanotubes (TNTs). This type of structure allows intercellular transport, not only for soluble small molecules but also for organelles such as lysosomes and mitochondria [41]. Studies on a mouse model using mesenchymal stem cells (MSC) as donor cells and airway epithelial cells as acceptor cells found that the overexpression of Miro-1 in MSCs enhanced the mitochondrial transfer and rescue of epithelial injury, while Miro-1 knockdown led to a loss of this ability. Moreover, allergic airway inflammation was reversed via Miro overexpression in allergen-induced asthma models [42]. Miro-1 regulates mitochondrial movement from stem cells to recipient epithelial cells and, thus, could have therapeutic potential. 

## 4. Miro GTPases in Microtubule Dynamics

Miro GTPases play important roles in regulating mitochondrial transport along microtubules, but do they also play roles in regulating microtubule dynamics? Microtubules are polar hollow tubes with a width of around 250 nm and feature a slow-growing minus end and a fast-growing plus end. The microtubules are attached to the microtubule organizing center (MTOC) through their minus ends, whereas the plus ends are oriented towards the periphery of the cell. Moreover, the plus ends undergo cycles of growth and shrinkage, a process known as dynamic instability, to meet the alternating needs of the cells, thereby changing their shapes and direction of migration [43]. Interestingly, there are several indications that Miro GTPases play more direct roles in regulating microtubule dynamics. One example is the observation that Miro GTPases can interact with the tumor suppresser protein Adenomatous Polyposis Coli (APC) [44]. APC is an important component in the Wnt signaling pathway. Mutations in the APC gene occur in around 80% of colorectal cancers and trigger the progression of sporadic colorectal cancers. APC is a multifaceted protein and was found to form bridges between all three types of skeletons: microtubules, actin filaments, and intermediate filaments [45]. One pool of APC accumulates at the microtubule ends and interacts with components of the +TIP complex such as end-binding protein 1 (EB1) (Figure 2). However, APC was also shown to localize to the mitochondria [46]. Silencing of APC in cancer cell lines resulted in perinuclear accumulation of mitochondria and reduced the number of motile mitochondria [43]. APC was, moreover, found to bind Miro-1 and TRAK2 and have a stabilizing function for the Miro-1/TRAK2 complex in the mitochondria. Cancer-associated truncated mutants of APC destabilized this interaction, leading to derailed mitochondrial distribution (Figure 2) [43]. In Drosophila, it was found that dMiro2 participates in properly positioning the APC-like protein Apc2 to dendrites [47].

Another example is the finding that Miro GTPases can bind histone deacetylase 6 (HDAC6), which is a microtubule-specific deacetylase (Figure 2) [48]. Microtubule acetylation is something that occurs on α-tubulin in microtubules with reduced dynamic instability and is a hallmark of stable microtubules. In turn, this phenomenon is associated with a reduced cell migratory index. HDAC6 negatively influenced axonal mitochondrial transport in rat neuronal cells, and HDAC6 inhibition resulted in increased transport, which was observed as accumulation in distant rat axons. Acetylated α-tubulin was not found to be the prime target for HDAC6. Instead, Miro-1 was found to mediate the HDAC6-dependent effects on mitochondrial transport [49]. Miro-1 can be acetylated on several lysine residues, but acetylated lysine 105 seems to be of particular importance as a target for HDAC6 [48]. Miro-1 deacetylation resulted in the decreased accumulation of mitochondria in rat axons and increased the responsiveness of Miro-1 to higher calcium concentrations. In contrast, HDAC6 inhibition or the expression of a Miro-1 mutant in position 105 mimicking lysine acetylation (Miro-1K105Q) increased mitochondrial transport, indicating that Miro-1 acetylation positively regulates Miro-1’s functions [49].

## 5. Miro GTPases Are Involved in Actin Dynamics

As previously noted, mitochondria mainly use microtubules for long-distance movement. For shorter and local transport, they can make use of the actin filament system. Mitochondrial trafficking along actin filaments cells was first demonstrated in chicken sympathetic neurons, finding that actin filament can provide a local transport system within axons [50]. Studies on Drosophila and C. elegans further demonstrated that actin-based mitochondrial trafficking involves myosin motor proteins, predominantly Myosin V and Myosin VI [51]. Importantly, the characterization of human Myosin 19, which is a member of the family of unconventional myosins, showed that this protein is specifically localized to the mitochondria through its tail domain. However, Myosin 19 was found not to be inserted into the outer mitochondrial membrane [52]. Instead, Myosin 19 was shown to bind Miro GTPases, which constitute the unifying link between the actomyosin motors and mitochondria [37]. Myo19 mitochondrial localization appears to require Miro GTPases, as Myo19 mitochondrial targeting is lost in cells expressing Miro GTPases lacking the C-terminal transmembrane domain [37].

It was postulated that mitochondria are needed in areas of a cell that have high demands for energy consumption. For instance, there is a significant accumulation of mitochondria at synapses and neuromuscular junctions [7,53,54]. In non-neuronal cells, such as fibroblasts and cancer cells, mitochondria target the leading edges of migrating cells. Although cancer cells are known to rely on glycolysis for ATP production, rather than oxidative phosphorylation, the formation of lamellipodia and membrane ruffles was found to require the local production of mitochondria-produced ATP (Figure 2) [55,56]. The intact capacity for mitochondria dynamics is vital, which can be illustrated by the observation that Drp1 is upregulated in invasive breast cancer cell lines, while the knockdown of Drp1 or overexpression of Mitofusin 1 inhibits lamellipodia formation [55]. Importantly, lamellipodia formation requires active actin polymerization and a dynamic reorganization of actin filaments into weave-like structures [57]. Active actin reorganization at the leading edge, fueled by ATP hydrolysis, increases the ADP:ATP ratio, which, in turn, leads to the activation of AMP-activated protein kinase (AMPK). Studies of human ovarian adenocarcinoma showed that AMPK can serve as a detector of reduced ATP concentrations, as its activity can regulate mitochondrial relocalization to the leading edges of migrating cells. Moreover, AMPK inhibition resulted in decreased lamellipodia formation and decreased cell migration [56]. Thus, the question remains: What is the role of Miro GTPases in mitochondrial localization to cell areas undergoing intense actin reorganization? Interestingly, studies on Miro-1-depleted mouse embryonic fibroblasts strongly suggested that Miro GTPases play a profound role in the localization of mitochondria to lamellipodia and, therefore, in lamellipodia activity, as Miro-1 ablation results in decreased collective cell migration [58]. Another link between Miro GTPases and actin regulators with an impact on tumorigenesis was demonstrated in studies on the relationship and possible direct interaction between the actin nucleation factor mDia2 and Miro-1 in cancer-associated fibroblasts (CAFs) [59]. The authors found that Miro-1 was upregulated in the stroma of liver, breast, and ovarian cancers and that Miro-1 and mDia2 co-expression in liver and lung cancers, which indicates increased migration activity, was associated with decreased patient survival. Together, mDia2 and Miro-1 seem to orchestrate the positioning of the mitochondria to areas of contact between CAFs and tumor cells [59].

If local ATP production is required for actomyosin-dependent cellular processes such as cell migration and cytokinesis, what role does GTP play? Notably, almost all proteins that participate in mitochondrial fusion, fission, and trafficking are GTP-binding proteins, which is also true for α- and β-tubulin dimers that comprise the building blocks of microtubules. In the cytoplasm, there is an excess of GTP over GDP by a factor of 10 [60]. However, the mechanism by which this ratio is maintained deserves further explanation. Most GTP is produced from two sources: through the activity of Nucleoside-diphosphate kinases (NDP kinases) and through the conversion of Succinyl-CoA to Succinate by Succinyl-CoA synthase (SCS) in the TCA cycle. As SCS is a mitochondrial enzyme, the mitochondria, which also contain NDP kinases, can likely regulate local GTP production, thereby providing fuel for the mitochondrial fusion/fission machinery and for the dynamic instability of microtubules.

## 6. Miro in Neuropathologies

Miro GTPases became associated with neuropathologies almost immediately after their discovery. Indeed, these GTPases play key roles in the regulation of mitochondrial trafficking, and numerous diseases such as Parkinson’s, Alzheimer’s, and amyotrophic lateral sclerosis (ALS) are associated with derailed mitochondrial functions [61,62,63,64]. Initially, this seemed to suggest guilt by association. However, the discovery that Miro GTPases directly bind to proteins identified in hereditary variants of Parkinson’s disease suggested more direct involvement. Familial forms of Parkinson’s disease account for around 5–10% of all cases, and mutations in two genes, PINK1 and Parkin, mediate autosomal recessive forms of the disease [65]. The confirmed interaction between Miro GTPases and PINK1 thus paved the way for the concept that Miro GTPases could have an influence on the progression of Parkinson’s disease, at least in the familial variants of the disease [66,67]. Miro GTPases were also shown to directly bind to the E3 ubiquitin ligase Parkin [68]. These discoveries sparked interest in elucidating the molecular interrelationships between these proteins and resulted in a deeper understanding of the molecular mechanisms underlying familiar Parkinsonism. The primary role of Miro GTPases in this context appears to involve the PINK1- and Parkin-dependent regulation of mitophagy, a process required for the clearance of dysfunctional mitochondria. Under healthy conditions, the PINK1 level is kept low via proteolytic degradation. In contrast, when mitochondria are damaged, the PINK1 level is stabilized, and the protein relocates to the outer mitochondrial membrane with the kinase domain facing the cytoplasmic side. In this position, PINK1 can phosphorylate Parkin, which leads to the activation of its E3 ubiquitin ligase. In addition, PINK1 can phosphorylate Miro GTPases, predominantly at serine 156, which, in turn, mediates the Parkin-dependent ubiquitination of Miro on several lysine residues, thereby targeting Miro for degradation. This scenario was demonstrated in several model systems, including Drosophila and mice [67,69,70,71]. Importantly, ubiquitination does not seem to involve the lysine residue 105. When acetylated, this residue is the target for HDAC6. In mouse embryonic fibroblasts, blocking Miro-1 ubiquitination (Miro-2 appears not to be involved in the process) stabilized Miro-1 protein levels and decreased mitophagy, indicating the central role of Miro-1 in the Parkin-mediated clearance of damaged mitochondria [71]. Notably, the expression of Miro-1P13V or Miro2A13V in cultured cells increased apoptosis, indicating that GTPase-defective Miro GTPases have a detrimental effect on mitochondria function [1].

Studies on human-induced pluripotent stem cells (iPSC) and an iPSC-derived neuronal model for familial and sporadic Parkinson’s disease demonstrated a direct interaction between Miro-1 and another Parkinson protein, Leucine-rich repeat kinase 2 (LRRK2) [72]. This interaction is important to allow for the degradation of Miro in response to mitochondrial stress as a proxy for Parkinson’s disease. A mutant LRRK2 found in PD patients (LRRK2G2019S) no longer interacted with Miro and was found to delay mitophagy [72]. Interestingly, reducing Miro-1 through RNA silencing in LRRK2G2019S-expressing human neurons and PD models of Drosophila rescued neurodegeneration, indicating that Miro-1 reduction might have a clinical impact on decreasing the neurodegenerative process [72]. Moreover, a study of skin fibroblasts from a large number of PD patients demonstrated that Miro-1 removal facilitated the clearance of defective mitochondria. Mitochondrial-stress-induced Miro-1 degradation was also found to be compromised in 94% of the fibroblasts [73]. Importantly, treatments with a small molecule predicted to target Miro-1 restored Miro-1 elimination, indicating that a reduction of Miro-1 could rescue some aspects of PD. These observations show that Miro-1 (but potentially not Miro-2) could serve as a target in PD treatment. Mutations in PINK1, Parkin, and LRRK2 have been found in familial and, in some cases, sporadic PD [65]. These findings have stimulated interest in discovering PD Miro mutants, but such mutations do not seem to be very common events. However, mutations in Miro-1, Miro-1R272Q, and Miro-1R450C were found in two PD patients. In these cases, the mutations decreased the number of ER–mitochondrial contact sites and derailed calcium homeostasis in fibroblasts obtained from mutation-carrying patients [74].

The roles of Miro GTPases in other neuropathologies are less clear. Many roles appear to be linked to the Parkin–PINK1–Miro axis of mitophagy regulation or the control of mitochondrial trafficking. One possible link between Miro GTPases and Alzheimer’s disease could be through a genetic interaction between Miro-1 and Tau, which is a protein implicated in Alzheimer’s disease [75,76]. Miro-1 has also been implicated in ALS, as it binds superoxide dismutase 1 (SOD1), which is a frequently mutated protein in familial ALS [77]. Expression of the common ALS-related mutant SOD1G93A in mouse motor neurons decreased both Miro-1 expression and axonal transport [78]. Miro-1 was also implicated in Huntington’s disease due to the expression of the disease-related protein 120QHtt in rat neurons’ decreased mitochondrial fusion, leading to mitochondrial shortening. Stimulating mitochondrial motility through overexpressing Miro-1 restored mitochondrial fusion rates and sizes under these experimental conditions [79]. 

## 7. Conclusions

Although the first publication on the subject indicated that Miro GTPases do not play the same active role in actin reorganization as conventional Rho GTPases, Miro GTPases were still found to have an impact on cytoskeletal dynamics. This role is primarily achieved through the targeting of mitochondria—and, thus, the ATP production machinery—in areas of the cell with high energy demands, such as the leading edges of migrating cells and the synapses in neuronal cells. Actin polymerization requires ATP, enabling Miro GTPases to participate in the redirection of mitochondria to cellular areas actively controlling the direction of cell migration (Figure 2). Miro GTPases are increasingly considered key components in protein complexes that operate at the intersection of mitochondrial dynamics and cytoskeletal reorganization. These processes must be tightly regulated to avoid the emergence of severe diseases, such as the neuropathological conditions mentioned above.

## Figures and Tables

**Figure 2 cells-13-00647-f002:**
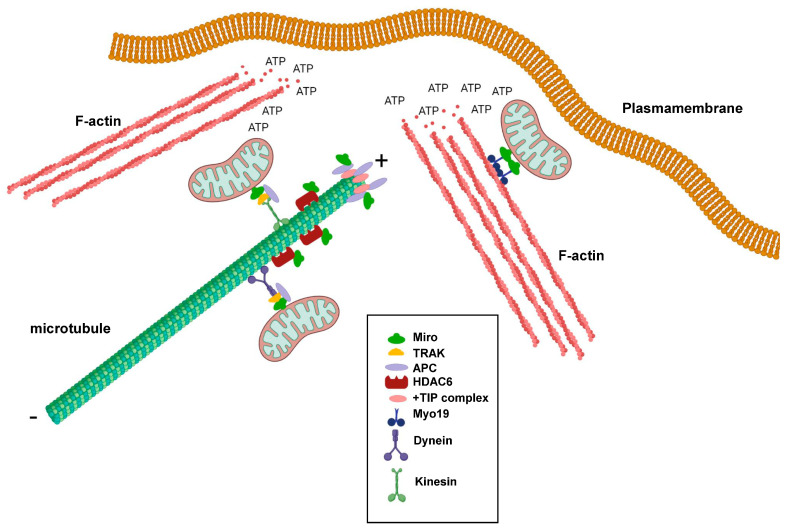
Schematic representation showing the involvement of Miro GTPases in regulating cytoskeletal dynamics and mitochondrial trafficking at the leading edge of migrating cells. + denotes the plus-end of microtubules and ‒ denotes the minus-end of microtubules. This figure was created with BioRender.com.

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
