# Peer review of "Miro GTPases at the Crossroads of Cytoskeletal Dynamics and Mitochondrial Trafficking"

_cells, 2024, doi:10.3390/cells13070647_

Round 1

Reviewer 1 Report

Comments and Suggestions for Authors

Main Remarks:

1.    The role of MiroGTPasas has previously been reviewed in neurodegenerative disorders. The author should include this review in his manuscript: Kay, L., Pienaar, I.S., Cooray, R. et al. Understanding Miro GTPases: Implications in the Treatment of Neurodegenerative Disorders. Mol Neurobiol 55, 7352–7365 (2018). https://doi.org/10.1007/s12035-018-0927-x

2.    The author should reorganize Figure 2.

Author Response

  1. This reference will be included in the revised manuscript.
  2. It is not clear how the Figure 2 should be reorganized. I am happy to revise the Figure if the reviewer could be more specific as to how it should be reorganized.

Reviewer 2 Report

Comments and Suggestions for Authors

The manuscript by Pontus Aspenstrom, entitled: “ Miro GTPases at the Crossroads of Cytoskeletal Dynamics and 2 Mitochondrial Trafficking”  reviews the discovery, structure, and function of the mitochondrial  Miro GTPases.  The review is comprehensive and detailed, as such it could prove a useful tool for new investigators in the field.  Before publication two items should be addressed that will greatly improve the quality of the work.

1.        The manuscript would greatly benefit from thoughtful editing by a native English speaker with a scientific background.  Many of the grammatical constructs are convoluted and unusual. 

2.        The conclusion in section 7 is not a conclusion/summary, instead, it primarily introduces and discusses  new concepts not previously discussed.  This new material would be better served moved to an earlier section, allowing this final section to summarize the content of the review and discuss its broader impact.   

Comments on the Quality of English Language

The manuscript would greatly benefit from thoughtful editing by a native English speaker with a scientific background.  Many of the grammatical constructs are convoluted and unusual. 

Author Response

  1. The manuscript has been edited thorough the English Editing service provided by MDPI.
  2. The new concept introduced in section 7 has been moved to section 5.

Reviewer 3 Report

Comments and Suggestions for Authors

Very interesting review.

Minor comments:

Line 8: … transport of mitochondria and peroxisomes transport along microtubules

Should be changed to … transport of mitochondria and peroxisomes along microtubules

Line 22: through advanced searches in the DNA sequence DNA-bases generated in the human genome project

May be changed to… through advanced searches in the DNA sequence generated in the human genome project

In Figure 1A: please revise the aligment of EF1, hEF1 and hEF2

Line 57: …through the interaction to the Parkinson proteins Parkin

May be changed to … through the interaction with the Parkinson proteins Parkin

Line 153: How, this is achieved at the molecular level is not entirely clear,

Should be changed to How this is achieved at the molecular level is not entirely clear,

Line 165: and resulted in an aggregation of mitochondria in the perinuclear area

Please, revise if aggregation is the more convenient term for the localization of mitochondria.

Line 246: Miro GTPases can interact with the tumor suppresser protein Adenomatous Polyposis Coli (APC)

Please, revise the format used for “Adenomatous Polyposis Coli” (it looks like it was written with a different size)

Line 283: and it was founds that actin filament can provide a local transport

Should be changed to “and it was found…”

Line 3017: …causes an increase in an increased ADP:ATP ratio

Should be changed to … causes an increase in ADP:ATP ratio

Line 359: iPSC-derived neuronal model för familial and sporadic Parkinson’s disease

Should be changed to … iPSC-derived neuronal model for familial and sporadic Parkinson’s disease

Line 375: This has, off course,

Should be changed to “This has, of course,”

Comments on the Quality of English Language

minor comments were suggested to the author

Author Response

I thank the reviewer for the positive comments! The text has revised according to the suggestions by the reviewer and Fig.1A has been corrected. In addition, the text has been through English Editing system provided by MDPI.